# Genetic and Epigenetic Profiles of Polycystic Ovarian Syndrome and In Vitro Bisphenol Exposure in a Human Granulosa Cell Model

**DOI:** 10.3390/biomedicines12010237

**Published:** 2024-01-21

**Authors:** Reem Sabry, Jenna F. Gallo, Charlie Rooney, Olivia L. M. Scandlan, Ola S. Davis, Shilpa Amin, Mehrnoosh Faghih, Megan Karnis, Michael S. Neal, Laura A. Favetta

**Affiliations:** 1Reproductive Health and Biotechnology Laboratory, Department of Biomedical Sciences, Ontario Veterinary College, University of Guelph, Guelph, ON N1G 2W1, Canada; rsabry@uoguelph.ca (R.S.); gallo@uoguelph.ca (J.F.G.); croone01@uoguelph.ca (C.R.); oscandla@uoguelph.ca (O.L.M.S.); odavis@uoguelph.ca (O.S.D.); 2ONE Fertility, Burlington, ON L7N 3T1, Canada; samin@onefertility.com (S.A.); mfaghih@onefertility.com (M.F.); mkarnis@onefertility.com (M.K.); mneal@onefertility.com (M.S.N.); 3Department of Obstetrics and Gynecology, McMaster University, Hamilton, ON L8S 4L8, Canada

**Keywords:** polycystic ovary syndrome, endocrine disrupting compounds, bisphenols, granulosa cells, microRNAs

## Abstract

Higher levels of bisphenols are found in granulosa cells of women with polycystic ovary syndrome (PCOS), posing the question: Is bisphenol exposure linked to PCOS pathophysiology? Human granulosa cells were obtained from women with and without PCOS, and genes and microRNAs associated with PCOS were investigated. The first phase compared healthy women and those with PCOS, revealing distinct patterns: PCOS subjects had lower *11β-HSD1* (*p* = 0.0217) and *CYP11A1* (*p* = 0.0114) levels and elevated miR-21 expression (*p* = 0.02535), elucidating the molecular landscape of PCOS, and emphasizing key players in its pathogenesis. The second phase focused on healthy women, examining the impact of bisphenols (BPA, BPS, BPF) on the same genes. Results revealed alterations in gene expression profiles, with BPS exposure increasing *11β-HSD1* (*p* = 0.02821) and miR-21 (*p* = 0.01515) expression, with the latest mirroring patterns in women with PCOS. BPA exposure led to elevated *androgen receptor (AR)* expression (*p* = 0.0298), while BPF exposure was associated with higher levels of miR-155. Of particular interest was the parallel epigenetic expression profile between BPS and PCOS, suggesting a potential link. These results contribute valuable insights into the nuanced impact of bisphenol exposure on granulosa cell genes, allowing the study to speculate potential shared mechanisms with the pathophysiology of PCOS.

## 1. Introduction

Polycystic ovary syndrome (PCOS) is a pervasive endocrine disorder that affects a substantial proportion of women in their reproductive years [1]. Defined by the triad of hyperandrogenism, anovulation, and polycystic ovarian morphology, PCOS intricately involves the malfunction of ovarian processes, leading to clinical manifestations that include irregular menstruation and infertility [1]. The characteristic cysts form when ovulation falters, and they begin to produce excess androgens, disrupting the delicate hormonal balance in women [2]. These androgens, typically present in minimal quantities, play a pivotal role in the regulation of menstruation and ovulation, implicating PCOS in female infertility [2].

Granulosa cells (GCs) are integral components of the cumulus–oocyte complex (COC) and encircle the developing oocyte to produce hormones and provide crucial support in fostering follicular development [3]. A normal menstrual cycle involves the orchestrated release of gonadotrophin-releasing hormone (GnRH), triggering the subsequent release of follicle-stimulating hormone (FSH) and luteinizing hormone (LH) from the pituitary gland [4]. In this intricate balance of hormones, LH stimulates theca cells to produce androgens, while FSH induces GCs to convert these androgens into estrogen [4]. Dysregulation of GC function in PCOS is evident through increased follicle numbers and GC proliferation, underscoring the critical role of GCs in the pathogenesis of the disorder [5].

The heightened pulsatility of GnRH coupled with elevated LH and diminished FSH levels sets the stage for PCOS [5]. The inability to convert androgens to estrogen and a follicle to fully mature characterizes the condition [5]. Most research has traditionally focused on the ovarian aspects of PCOS, yet the pivotal role of GCs in steroidogenesis necessitates a nuanced examination of their association with the disorder [6]. Though the etiology of PCOS remains unclear, a combination of factors including genetic predisposition and environmental influences are believed to play a role in its development [6]. Key genes, including the *androgen receptor (AR)* gene and hydroxysteroid dehydrogenase (HSD) enzymes, are central to the investigation [7]. The *AR* gene, expressed in oocytes, GCs, and theca cells, produces the androgen receptor protein, with PCOS-afflicted individuals exhibiting heightened AR activity [7]. HSD enzymes, particularly 11β-hydroxysteroid dehydrogenases (11β-HSDs), are crucial players in steroid biosynthesis and metabolism [8]. The dysregulation of 11β-HSD1 and the ambiguous role of 11β-HSD2 in PCOS underscore the complex interplay between hormonal regulation and the disorder [9]. *Cytochrome P450 family 11 subfamily A member 1 (CYP11A1)* and *cytochrome P450 family 17 subfamily A member 1 (CYP17A1)* genes add another layer of complexity, expressed in the adrenal glands and gonads, these genes contribute to steroidogenesis [10]. Polymorphisms in these genes emerge as risk molecular markers for PCOS, emphasizing the complicated genetic landscape of the disorder [11].

Beyond the realm of genetics, investigating the epigenetic profile associated with PCOS is of great interest to researchers to gain a better understanding of the pathophysiology behind this disease. MicroRNAs (miRNAs), small non-coding RNAs pivotal in post-transcriptional gene expression, introduce another avenue to explore the expression landscape of PCOS [12]. Research highlights the differential expression of circulating miRNAs in PCOS-diagnosed women and has hinted at their potential use as both diagnostic and therapeutic biomarkers [12]. Several miRNAs are differentially expressed in GCs of women with PCOS, and this study further aimed to explore key microRNAs crucial for GC steroidogenesis. These include miR-21, miR-34c, and miR-155 emerging as key players in GC function and follicular development [12,13,14].

In contrast to intrinsic genetic and epigenetic factors, endocrine-disrupting compounds (EDCs) are also known to increase the risk of developing this disorder [15]. Bisphenols, including BPA, BPS, and BPF, particularly garner attention for their potential to influence a variety of reproductive and endocrine disorders, including PCOS [15]. Despite the government limitations on the use of BPA in manufacturing due to health concerns, traces persist in the population, with women diagnosed with PCOS exhibiting elevated levels of BPA in their urine [16]. The analogs, BPS and BPF, gaining popularity, pose an uncertain long-term health risk, prompting the need for continued research [17].

Considering the high prevalence of PCOS and its implications for women’s health, delving into the complex web of associations between environmental influences, specifically bisphenols, and the pathogenesis of PCOS is of importance. Women with PCOS face an elevated risk of associated disorders, such as type II and gestational diabetes, high blood pressure, heart disease, stroke, sleep apnea, and endometrial cancers, amplifying the urgency for a comprehensive understanding of the disorder [18]. This study, therefore, seeks to unravel the potential mechanistic links between bisphenol exposure and PCOS pathophysiology. The hypothesis suggests a correlation between bisphenols and PCOS in human granulosa cells (hGCs), postulating shared genetic and epigenetic profiles between PCOS and bisphenol-exposed cells.

The selection of genes aimed to investigate genes and miRNAs associated with steroidogenesis, since this is a crucial component of hormonal regulation, particularly in the context of PCOS. Since PCOS is associated with disruptions in hormonal balance, the chosen genes and miRNAs were selected based on their participation in steroid hormone synthesis and their potential implications in PCOS pathogenesis. The AR is a pivotal player in mediating the effects of androgens, which are known to be overproduced in patients with PCOS [7]. The cytochrome genes including *CYP17A1* and *CYP11A1* are both involved in hormone biosynthesis within granulosa cells and are shown to be dysregulated in women with PCOS [10]. The hydroxysteroid dehydrogenases regulate cortisol levels and are metabolically linked to PCOS processes [9]. miR-21 is one of the most investigated microRNAs and is reported to be affected by bisphenols in numerous studies, including several from our research group [12]. Increases in miR-21 have been linked to androgen excess and PCOS. miR-155 is involved in inflammatory aspects of PCOS, and miR-34c is linked to altered cell dynamics observed in PCOS [12,13,14].

## 2. Materials and Methods

### 2.1. Ethics Approval and Patient Criteria

Ethics approval to collect the human granulosa cells (hGCs) as biomedical waste was obtained from the Hamilton Integrated Research Ethics Board (HiREB) under project #11-252-T in May 2021 and was extended to the University of Guelph Collaborators (Dr Favetta) on 1 September 2021.

Women undergoing controlled ovarian stimulation (COH) using recombinant follicle-stimulating hormone (rFSH) for fertility treatment (IVF) between the ages of 25 and 40 were included in the study. Women undergoing treatment for fertility preservation were excluded.

### 2.2. Cell Retrieval

Human granulosa cells (hGCs) were obtained from ONE Fertility in Burlington, ON, Canada from patients undergoing in vitro fertilization (IVF) treatments during the egg retrieval stage. Granulosa cells surrounding the oocytes were dissected from the COC and isolated in Eppendorf tubes containing 2 mL of fresh DMEM/F12 collection media. The tubes were held at 4 °C during transport to the University of Guelph at the Reproductive Health and Biotechnology laboratory. Upon arrival, cells were isolated, washed in PBS, and treated with a hemolysis buffer. Initially, cells from control and PCOS women were snap-frozen in liquid nitrogen and stored in a −80 °C freezer for subsequent RNA extraction and qPCR analysis. The second part of the study included cells obtained from control women only, with no PCOS diagnosis. Following the PBS wash, these cells were plated on a T25 flask with DMEM/F12 containing 20% FBS and incubated at 37.5 °C with 5% CO_2_ for 8 days with fresh media replacement every 48 h until flasks were confluent. Next, cells were passaged once before being exposed to bisphenols.

### 2.3. Cell Culture and Bisphenol Treatment

Passage 1 human granulosa cells (hGCs) were split into five wells on 6-well plates at a seeding density of 1 × 10^4^ cells/well in DMEM + 10% FBS. The cells were incubated at 37.5 °C with 5% CO_2_ for 24 h before serum starvation. The media was replaced with OptiMEM Serum Restricted Media for another 24 h before bisphenol treatment. The 5 wells were separated into control, vehicle, BPA, BPS, and BPF. The vehicle was treated with 0.1% ethanol to mimic how the BPA was dissolved into solution. Next, the respective wells were treated with either BPA, BPS, or BPF at a dose of 0.05 mg/mL and incubated for another 24 h. After treatment, the cells were snap-frozen in liquid nitrogen and stored at −80 °C for downstream RNA analysis.

To establish significant experimental doses for human granulosa cells, it is crucial to consider the lowest observed adverse effect level (LOAEL). Previous research on bisphenol A (BPA) indicated the LOAEL to be 50 mg/kg/day for in vivo studies, and when translated to in vitro doses, it was calculated as 50 μg/mL [19]. Additionally, preliminary ELISA experiments conducted by our group showed that oocytes treated with this LOAEL dose of BPA (50 μg/mL) had a BPA uptake in the same range of the levels measured in human follicular fluid (2.4 ng/mL) as reported by Ikezuki et al. [20]. This correlation further supports the use of the in vitro LOAEL dose as the relevant and physiologically significant dose for investigating the effects of bisphenols on human granulosa cells in culture. Our group previously confirmed this to be the optimal dose to use by dose–response experiments and published several articles in diverse journals with the use of this specific dose.

The selection of a 24 h treatment duration was motivated by time-dependent experiments previously conducted in our laboratory and by the literature. The authors qualitatively monitored the cells after treatment at intervals of 12 h to closely observe any increase in cell death. Furthermore, a previous study conducted by Mansur et al. [21] used a 48 h treatment duration with a lower dose of 20 µg/mL. However, in our previously conducted experiments, granulosa cells died when exposed to bisphenols beyond 24 h. Considering this, the authors reduced the treatment duration to 24 h to maintain cell viability, while still capturing BPA potential adverse effects.

### 2.4. RNA Extraction and cDNA Synthesis

The Qiagen miRNeasy Micro Kit (Qiagen, Toronto, ON, Canada) was used to purify and extract total RNA as per the manufacturer’s protocol. Briefly, frozen cells were treated with a QIAzol Lysis Reagent followed by chloroform treatment. The upper aqueous phase was collected and placed into a RNeasy MinElute spin column to wash and purify the total RNA. The column was then dried and subjected to RNase-free water to elute the RNA. RNA concentration and quality were measured using the Nanodrop 2000c (ThermoFisher, Whitby, ON, Canada). Two hundred nanograms of mRNAs and miRNAs were reverse transcribed using qScript complementary DNA (cDNA) Supermix and qScript microRNA cDNA Synthesis kit, respectively, in a T100 Thermal Cycler. cDNA was diluted using RNase-free water to a concentration of 5 ng/μL for mRNAs and 1.5 ng/μL for miRNA prior to qPCR.

### 2.5. Quantitative Polymerase Chain Reaction (qPCR)

The levels of mRNA and miRNA expression were measured using quantitative real-time PCR (qPCR) with the CFX96 Touch Real-Time PCR Detection System from BioRad (Mississauga, ON, Canada). Amplification of mRNAs was performed with the SsoFast EvaGreen Supermix, and miRNAs were amplified using the PerfeCTa SYBR Green Supermix. mRNA and miRNA primers were purchased from Sigma-Aldrich (Oakville, ON, Canada) and Qiagen (Toronto, ON, Canada), respectively. All primers were tested using standard curves with efficiencies accepted only with values between 90 and 110%. Gene expression was calculated using the efficiency-corrected method (ΔΔCt). Primer sequences and efficiencies are given in Table 1 and Table 2. mRNA expression was normalized to housekeeping genes *Tyrosine 3-monooxygenase/tryptophan 5-monooxygenase activation protein zeta* (YWHAZ) and *Ribosomal Protein Lateral Stalk Subunit P0 (RPLP0)*, as they were determined to be the most stable reference genes according to a GeNorm analysis using the CFX Maestro Software 2.3 (Appendix A). miRNA expression was normalized to miR-191 and miR-106a, as they are stable reference targets across treatments [22]. All quantification was run on at least three biological replicates in technical triplicates. miRNA PCR signal acquisition was carried out using the following three-step PCR cycling protocol: 95 °C for 2 min followed by 39 cycles of 95 °C for 5 s, 60 °C for 30 s, 70 °C for 30 s, ending with melt curve acquisition from 60 to 95 °C. mRNA PCR signal acquisition was carried out using the following two-step PCR cycling protocol: 95 °C for 2 min followed by 44 cycles of 95 °C for 10 s, 60 °C for 30 s, ending with melt curve acquisition from 60 to 95 °C.

### 2.6. Statistical Analysis

Statistical analyses were conducted using GraphPad Prism 6 software. Differences in expression levels between normal cells treated with bisphenols were analyzed using normality tests. The normality of each data set was assessed with the Shapiro–Wilk test. For normally distributed data, a one-way analysis of variance (ANOVA) was applied, while non-parametrically distributed data sets were analyzed using the Kruskal–Wallis test. Significance was determined at a two-tailed *p*-value of ≤0.05. Post hoc tests, namely, Tukey’s for parametric data and Dunn’s multiple comparison tests for non-parametric data, were employed on sets with a statistically significant *p*-value to assess differences between individual treatment groups. Differences in expression levels between normal and PCOS cells were analyzed using an unpaired Student’s *t*-test. The presented data represent the mean ± standard error of the mean (SEM) for biological replicates, and statistical significance was established at a two-tailed *p*-value of ≤0.05; thus, any differences with *p* ≤ 0.05 were considered significant unless stated otherwise in the figure legends.

## 3. Results

### 3.1. Gene Expression Profiles Differ in Women with PCOS and Control Women

hGCs from women with PCOS and controls were snap-frozen, RNA was extracted, reverse-transcribed, and relative mRNA/miRNA expression was quantified using qPCR. Both *11β-HSD1* and *CYP11A1* were significantly lower in women with PCOS than in the controls (*p* < 0.05) as shown in Figure 1A,C. PCOS-affected cells and controls showed no difference in relative mRNA expression levels for all other genes investigated including *11β-HSD2*, *CYP17A1*, and *AR* (Figure 1B,D,E). miR-21 was the only miRNA that was significantly differentially expressed between control and PCOS-affected granulosa cells with higher levels seen in GCs from women with the disorder (*p* < 0.05) (Figure 2A). miR-155 and miR-34c were unchanged between our two groups (Figure 2B,C).

### 3.2. BPA and Analogs Disrupt Normal Gene Expression of Genes Associated with PCOS

Human granulosa cells (hCGs) from healthy patients only were cultured and treated with a vehicle (ethanol) or bisphenols (BPA, BPS, BPF) at the currently reported lowest-observed-adverse-effect level (LOAEL) dose (0.05 mg/mL) for BPA [22]. RNA analysis was performed as described above. Figure 3 and Figure 4 represent only control patients treated with either vehicle or bisphenols. The *androgen receptor (AR)* was significantly increased in GCs treated with BPA compared to the control (*p* < 0.05) (Figure 3E). The same is true for *11β-HSD1*, which showed a significant increase after treatment with BPS (*p* < 0.05) (Figure 3A). In regard to the miRNAs, both miR-21 and miR-155 were affected by bisphenol exposure (Figure 4). However, miR-21 was vulnerable to BPS with a significant increase after exposure (Figure 4A), whereas miR-155 exhibited that same significant increase after exposure to BPF (Figure 4B). miR-34c was unaffected by all the bisphenols tested.

## 4. Discussion

Polycystic ovarian syndrome (PCOS) is an intricate endocrine disorder linked to reproductive, metabolic, and hormonal dysregulations. Our study aimed to decipher the molecular intricacies associated with PCOS by examining the expression profiles of key genes (*11B-HSD1*, *11B-HSD2*, *CYP17A1*, *CYP11A1*, *AR*) and microRNAs (miR-21, miR-155, miR-34c) in human granulosa cells. Additionally, we explored how exposure to bisphenols (BPA, BPS, BPF) might induce similar alterations, shedding light on potential environmental contributors to PCOS and further elucidating the intricate relationships between gene expression and endocrine disruptors.

In our investigation of women with PCOS, a striking downregulation of *11β-HSD1* and *CYP11A1* was observed, suggesting disruptions in androgen biosynthesis and metabolism. This finding aligns with studies implicating these genes in PCOS pathogenesis. Michael et al. [9] reported diminished 11β-HSD1 levels in women with PCOS. They suggested that the increased presence of androgens has an inhibitory role on this gene. Furthermore, this may contribute to the decreased inactivation of cortisol in follicles leading to a block of folliculogenesis [9]. Furthermore, increased levels of CYP11A1 are also correlated with hyperandrogenism and PCOS [30,31]. CYP11A1 is the key enzyme in the cholesterol metabolism to pregnenolone that eventually gets converted into progesterone [31]. Disruption of this gene will undoubtedly interrupt normal steroidogenesis and contribute to the pathogenesis of PCOS.

Lastly, miR-21 is one of the miRNAs we investigated in this study, as it is one of the most important miRNAs in granulosa cell function and has been correlated with PCOS and EDC exposure in numerous studies [22,32,33,34]. Elevated miR-21 levels in GCs of PCOS patients contribute significantly to follicular health and survival, playing a pivotal role in PCOS pathophysiology and follicular dysfunction [35]. miR-21, identified in bovine, ovine, and human granulosa cells, is typically increased during folliculogenesis, impacting the growth of preovulatory follicles [36].

The influence of miR-21 extends to its target genes, including bone morphogenetic protein receptor type II (BMPR2), essential for oocyte–somatic cell communication. BMPR2 activation influences granulosa cell proliferation, BMP15 and GDF9 signaling, and disruptions in this pathway can lead to infertility and reproductive abnormalities [37]. Additionally, miR-21 regulates pentraxin-3 (PTX3), critical for granulosa cumulus cell expansion. Decreased PTX3 expression is associated with impaired ovulation in mice, highlighting its importance in oocyte development [38]. Thus, the upregulation of miR-21 in PCOS patient GCs intricately regulates *BMPR2*, *PTX3*, and several other genes, impacting oocyte maturation and early embryo development. In women with PCOS, all these pathways may be disrupted due to higher levels of miR-21, which is a contributing factor to the development of this condition.

Turning our attention to the effects of bisphenol exposure on healthy women’s granulosa cells, nuanced alterations in gene and microRNA expression emerged. BPS exposure resulted in the upregulation of *11β-HSD1* and miR-21, potentially mirroring androgen metabolism dysregulation akin to PCOS. BPS exposure may potentially contribute to the onset of PCOS in women; this is supported by the identical gene expression profiles, where the same genes are affected in the same way (*11β-HSD1* and miR-21), while the rest of the genes investigated were unaffected by BPS and were not differentially expressed in women with PCOS. To the best of our knowledge, no studies have explored the effects of this BPA analog on *11β-HSD1*. Previous studies in bovine granulosa cells reported that BPS did not affect miR-21 expression [22], suggesting that human cells may be more vulnerable to the effects of BPS. Furthermore, BPA exposure induced increased *AR* expression, consistent with Richter et al. [39], while BPF exposure led to elevated miR-155 expression, in concordance with the findings of Oldenburg et al. [40].

Interestingly, Oldenburg et al. [40] reported the opposite effect on miR-155 when higher doses of BPF were used. They explained the non-monotonous dose response that is typically seen with bisphenols; this challenges the conventional toxicological paradigm and further complicates our understanding of bisphenol effects. The diverse responses to BPA, BPS, and BPF underscore the intricate relationship between bisphenol structure and its impact on gene and microRNA regulation. Winkler et al. [41] highlighted the importance of structural variations, demonstrating how subtle changes can result in differential modulation of gene expression. The different structures provide each bisphenol with unique characteristics that can explain why they did not have the same effect on our genes in this study. Researchers have provided insights into the potential mechanisms behind bisphenol-induced endocrine disruption, suggesting its role in PCOS pathogenesis.

Bisphenol A (BPA) and its analogs significantly influence pathways like insulin signaling, lipid metabolism, ovarian steroidogenesis, and the hypothalamic–pituitary–gonadal axis (HPG), as observed in animal models and human cell line studies [42,43]. BPA, mimicking estrogen, disrupts steroid feedback at the hypothalamus–pituitary level and ovarian steroid action, altering the HPG axis [42]. This includes disruption of LH and FSH secretions, which can contribute to PCOS development in premenopausal women [42]. BPA also contributes to metabolic and endocrine disorders in PCOS by promoting insulin resistance, inflammation, and hyperandrogenism [44]. It affects aromatase expression, causing dysregulation in estrogen production and potentially inducing hyperandrogenism [17]. The persistence of incompletely metabolized BPA disrupts gene expression, affecting gonadotropin secretion, ovarian steroidogenesis, and insulin activity, contributing to the clinical manifestations of PCOS [45].

This study possesses various strengths and limitations that were taken into consideration when interpreting the results. For starters, this study addresses clinically relevant questions of whether bisphenol exposure is linked to the pathophysiology of PCOS and provides insights into potential mechanisms underlying the condition. Furthermore, the utilization of human granulosa cells from women undergoing IVF treatments is a non-invasive approach to investigating these mechanisms and enhances the clinical relevance of the findings, offering a direct link to reproductive health. Lastly, the identification of a parallel epigenetic expression profile between BPS and PCOS offers valuable insights into epigenetic aspects of bisphenol exposure in the context of PCOS. On the other hand, the study acknowledges its limitations including the small sample size of patients with PCOS. Due to decreased consent and lower encounters of PCOS patients, only a small sample size was obtained for the study. Furthermore, the study primarily relies on observational data, limiting the ability to establish causation between bisphenol exposure and PCOS. Further directions include culturing granulosa cells from patients with PCOS and treating them with bisphenols to strengthen causal relationships.

Our study adds to the growing body of evidence suggesting that bisphenol exposure may induce molecular alterations reminiscent of those seen in PCOS, but it is essential to acknowledge the complexity of these interactions. This study only lays down the basic body of information upon which future research should build and delve into the intricate molecular mechanisms underpinning bisphenol-induced alterations, exploring the multifaceted interplay between these compounds and endocrine regulation. Additionally, elucidating the long-term consequences of bisphenol exposure on women with PCOS requires extensive investigation. In conclusion, our study provides a comprehensive examination of the complex interplay between PCOS, bisphenols, and the genetic and microRNA landscape of human granulosa cells, underscoring the need for continued research to unravel the intricate mechanisms linking environmental exposures to reproductive disorders.

## Figures and Tables

**Figure 1 biomedicines-12-00237-f001:**
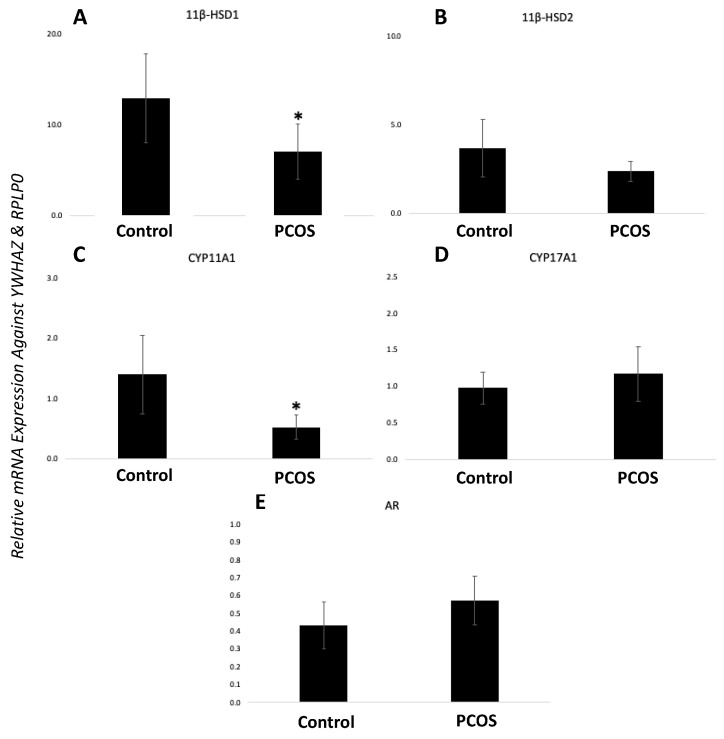
Gene expression profiles in GCs from women with PCOS vs. control women. Human granulosa cells were snap-frozen in liquid nitrogen from either healthy patients or patients with PCOS. RNA was quantified using qPCR and normalized against *YWHAZ* and *RPLP0:* (**A**) *11β-HSD1*, (**B**) *11β-HSD2*, (**C**) *CYP11A1*, (**D**) *CYP17A1*, and (**E**) *AR* genes. Bars represent mean ± SEM, *n* = 6, * *p* < 0.05. All data sets were normally distributed and were analyzed using an unpaired student’s *t*-test.

**Figure 2 biomedicines-12-00237-f002:**
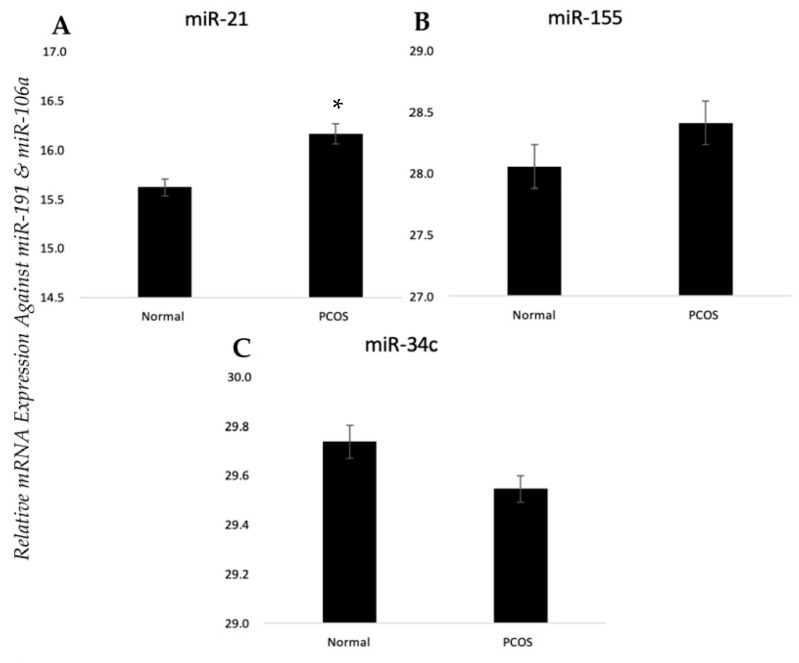
miRNA expression profiles in GCs from women with PCOS vs. control women. Human granulosa cells were snap-frozen in liquid nitrogen from either healthy patients or patients with PCOS. miRNAs were quantified using qPCR and normalized against miR-191 and miR-106a. (**A**) miR-21 was the only differentially expressed among them, and (**B**) miR-155 and (**C**) miR-34c were unaffected. Bars represent mean ± SEM, *n* = 3, * *p* < 0.05. This normal data set was analyzed using an unpaired student’s *t*-test.

**Figure 3 biomedicines-12-00237-f003:**
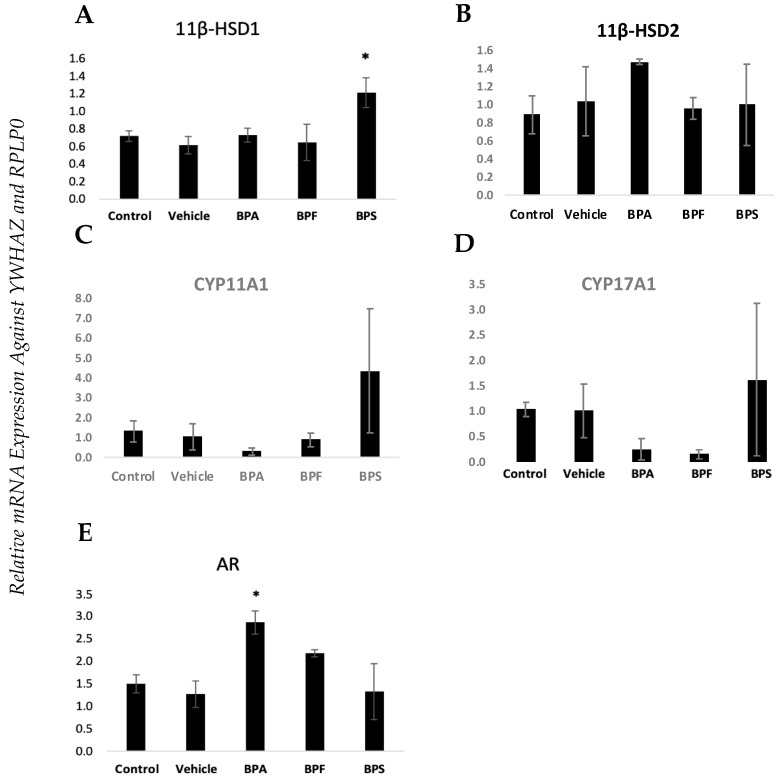
Gene expression profiles in cultured GCs exposed to bisphenols. mRNA expression of bisphenol-treated human granulosa cells of (**A**) *11β-HSD1*, (**B**) *11β-HSD2*, (**C**) *CYP11A1*, (**D**) *CYP17A1*, and (**E**) *AR genes*. Human granulosa cells were cultured in vitro and treated with either ethanol (vehicle) or BPA, BPS, and BPF at 0.05 mg/mL for 24 h. RNA was quantified using qPCR and normalized against *YWHAW* and *RPLP0*. This data set was analyzed using a one-way ANOVA for normally distributed data sets (*AR* and *11β-HSD1*) and using a Kruskal–Wallis test for non-normal data sets (*11β-HSD2*, *CYP17A1*, and *CYP11A1*). Bars represent mean ± SEM, *n* = 4, * *p* < 0.05 for normally distributed data and median ± MED (median absolute deviation), *n* = 4, * *p* < 0.05 for non-normally distributed.

**Figure 4 biomedicines-12-00237-f004:**
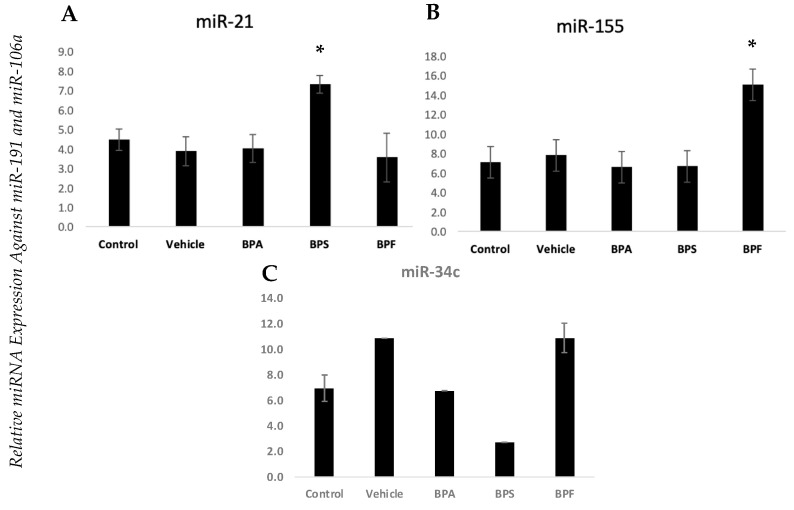
miRNA expression profiles in cultured GCs exposed to bisphenols. miRNA expression of bisphenol-treated human granulosa cells of (**A**) miR-21, (**B**) miR-155, and (**C**) miR-34c. Human granulosa cells were cultured in vitro and treated with either ethanol (vehicle) or BPA, BPS, and BPF at 0.05 mg/mL for 24 h. miRNA was quantified using qPCR and normalized against miR-191 and miR-106a. This data set was analyzed using a one-way ANOVA for normally distributed data sets (miR-21 and miR-155) and using a Kruskal–Wallis test for non-normal data sets (miR-34c). Bars represent mean ± SEM, *n* = 3, * *p* < 0.05 for normally distributed data and median ± MED, *n* = 3, * *p* < 0.05 for non-normally distributed.

**Table 1 biomedicines-12-00237-t001:** miRNA primer sequences for qPCR.

MicroRNA	Primer ID	Accession #	Sequence (5′-3′)	E (%)	Source
miR-191	hsa-miR-191-5p	MIMAT0000440	AACGGAATCCCAAAAGCAG	99.7	[23]
miR-106a	hsa-miR-106a-5p	MIMAT0000103	CGCCAAAAGTGCTTACAGTGC	92.4	
miR-21	bta-miR-21-5p	MIMAT0003528	TAGCTTATCAGACTGATGTTGACT	96.7	[24]
miR-34c	bta-miR-34c	MIMAT0003854	AGGCAGTGTAGTTAGCTGATTGC	99.6	[25]
miR-155	hsa-miR-155-5p	MIMAT0000646	TGCTAATCGTGATAGGGGTAAA	100	[26]

**Table 2 biomedicines-12-00237-t002:** mRNA primer sequences for qPCR.

Gene Symbol	Gene Name	Product Size (bp)	Accession #	Primer Sequence Sets (5′-3′)	E (%)	Source
YWHAZ	* Tyrosine 3-monooxygenase/tryptophan 5-monooxygenase activation protein zeta *	245	NM_001135699.1	F: ACTTTTGGTACATTGTGGCTTCAA R: CCGCCAGGACAAACCAGTAT	100.1	[27]
RPLP0	* Ribosomal protein lateral stalk subunit P0 *	240	NM_001002.3	F: AGCCCAGAACACTGGTCTC R: ACTCAGGATTTCAATGGTGCC	100.7	
11β-HSD1	*11 Beta-hydroxysteroid dehydrogenase type I*	180	NM:001,206,741.1	F: GCATTGTTGTCGTCTCCTCT R: TGGCTGTTTCTGTGTCTATGAG	100.9	[28]
11β-HSD2	*11 Beta-hydroxysteroid dehydrogenase type 2*	162	NM:000,196.3	F: GCTGTGAACTCCTTCCCT R: CGATGTAGTCCTTGCCGT	99.3	
CYP17A1	*Cytochrome P450 17A1*	154	NM:000,102.3	F: GATAACCACATTCTCACCACC R: GGCTGAAACCCACATTCTG	100.9	
CYP11A1	*Cytochrome P450 11A1*	169	NM:000,781.2	F: CTTCCTTTCTGTCTCAATTCCC R: TCTACCAGATGTTCCACACC	100.3	
AR	*Androgen receptor*	155	NM_000044.6	F: GCCTTGCTCTCTAGCCTCAA R: GGTCGTCCACGTGTAAGTTG	100.9	[29]

## Data Availability

The data presented in this study are available on request from the corresponding author.

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
