# Peer review of "Genetic and Epigenetic Profiles of Polycystic Ovarian Syndrome and In Vitro Bisphenol Exposure in a Human Granulosa Cell Model"

_biomedicines, 2024, doi:10.3390/biomedicines12010237_

Round 1
Reviewer 1 Report
Comments and Suggestions for Authors
In the current study the authors aimed to assess the putative correlation between bisphenols exposure and PCOS focusing on the genetic and epigenetic profiles in human granulosa cells. The manuscript is clear and well written. However, it can be improved according to the comments below.
1. Please provide a clinical descriptio of the enrolled patients, including inclusion and exclusion criteria
2. Please provide the rationale of using a specific protocol (dose and timing) for CF treatment. Did the authors performed preminary studies on that? Provide reference, if any.
3. Please clarify the reason for the selection of specific target genes and miRNAs
4. It is not clear wheter figure 3 and 4 are related to PCOS or control patients. Please clarify and add a comparison between the two groups
Comments on the Quality of English Language
Minor editing of English language required
Author Response
Please, see attached pdf file

Reviewer 2 Report
Comments and Suggestions for Authors
This is a well-designed and interesting study.
Some suggestions to improve the quality of the paper:
1. In the abstract section, one sentence of introduction (background: what is known) and some information about the methods should be included. Not only the aim and results of the study.
2. "Endocrine-disrupting compounds" should be added to keywords.
3. The second part of the study included cells obtained from control women only, with no PCOS diagnosis. – where the control group was obtained? Were there exclusion and inclusion criteria for this group?
4. Rnase-free – please, capitalize the second letter (all main txt)
5. Why does the author present results only as the means? The median should be present if the data had a non-normal distribution.
6. In the Figures, authors should provide information about statistical tests used to examine group differences.
7. The strengths and limitations of the study should be mentioned.
8. Please add in vitro studies which aim to assess the relationship between bisphenols and PCOs, such as
https://www.sciencedirect.com/science/article/pii/S004565352203956X
https://www.sciencedirect.com/science/article/abs/pii/S0890623823001855
https://link.springer.com/article/10.1007/s11356-023-26820-w
and discuss this results.
Author Response
Please, see attached pdf file

Round 2
Reviewer 1 Report
Comments and Suggestions for Authors
The authors have addressed all my comments